# Age-Related Glucose Intolerance Is Associated with Impaired Insulin Secretion in Community-Dwelling Japanese Adults: The Kumamoto Koshi Study

**DOI:** 10.3390/biomedicines13020380

**Published:** 2025-02-06

**Authors:** Kazuki Fukuda, Masaki Haneda, Naoto Kubota, Eiichi Araki, Kazuya Yamagata

**Affiliations:** 1Center for Metabolic Regulation of Healthy Aging (CMHA), Faculty of Life Sciences, Kumamoto University, Kumamoto 860-8556, Japan; fudakazuki@kumamoto-u.ac.jp (K.F.); nkubota0511@kumamoto-u.ac.jp (N.K.); earaki@gpo.kumamoto-u.ac.jp (E.A.); 2Department of Metabolic Medicine, Faculty of Life Sciences, Kumamoto University, Kumamoto 860-8556, Japan; msk_asp72@yahoo.co.jp; 3Research Center for Health and Sports Sciences, Kumamoto Health Science University, Kumamoto 861-5598, Japan; 4Diabetes Center, Kikuchi Medical Association Hospital, Kumamoto 861-1306, Japan; 5Department of Medical Biochemistry, Faculty of Life Sciences, Kumamoto University, Kumamoto 860-8556, Japan

**Keywords:** age, elderly, insulin secretion, insulin sensitivity

## Abstract

**Background/Objectives**: Glucose tolerance progressively declines with age. However, the effects of aging on insulin secretion and insulin sensitivity in Japanese subjects are unclear. **Methods**: We conducted an oral glucose tolerance test (OGTT) in residents aged between 22 and 85 years in Koshi City, Kumamoto Prefecture, Japan, to clarify the characteristics of insulin secretion and insulin sensitivity in older adults. Participants were recruited using a flyer, and the OGTT was performed after an overnight fast (12–16 h) between 8:00 and 10:30 am. **Results**: HOMA-IR and the Matsuda index are indices of insulin action. No correlation of age with HOMA-IR or the Matsuda index was found, whereas HOMA-β, the insulinogenic index, and the disposition index, all indices of insulin secretion, were negatively correlated with age in all participants and in individuals with normal glucose tolerance. Multiple regression analysis showed that age was an explanatory factor for insulin secretion. **Conclusions**: Impaired insulin secretion may contribute to age-related glucose intolerance in Japanese individuals.

## 1. Introduction

Glucose tolerance progressively declines with age [1,2]. Older people are more glucose intolerant than young individuals, and the total diabetes prevalence exceeds 20% in every age group between 65 and 95 years worldwide [3]. Insulin resistance and impaired insulin secretion are the major underlying mechanisms of hyperglycemia in type 2 diabetes. The effects of aging on insulin action and insulin secretion have been studied in detail, but the results are controversial. Regarding insulin action, many studies indicate that insulin resistance develops with age [4,5,6], but some studies did not find such an association [7,8,9]. The effect of aging on insulin secretion is also unclear. Previous studies have been inconsistent, reporting that aging increases [6,10], decreases [4,7,11,12,13], or does not change [5,14,15] insulin secretion. This variability may be due to confounding factors associated with aging, such as obesity and insulin resistance [1].

Ethnic differences in insulin sensitivity and insulin secretion are widely recognized, and Japanese people are characterized by higher insulin sensitivity and lower insulin secretion [16,17,18]. Controversial results have also been reported on the effect of aging on insulin sensitivity and insulin secretion in Japanese people. One study reported that both the homeostasis model assessment of insulin resistance (HOMA-IR), an index of insulin resistance; and the homeostasis model assessment of β-cell function (HOMA-β), a marker of β-cell function; declined with age [19], suggesting that insulin secretion decreases with age while insulin sensitivity increases. On the other hand, another study showed an age-dependent deterioration of insulin sensitivity using the Matsuda index, but no age-dependent change in the insulinogenic index, a marker of glucose-stimulated early insulin secretion, in Japanese subjects after the age of 65 [20]. Insulin secretion in an oral glucose tolerance test (OGTT) was also comparable between middle-aged and elderly Japanese individuals [21]. Therefore, the effect of aging on insulin sensitivity and insulin secretion in Japanese people remains unclear.

Characterizing insulin secretion and insulin action in older people would be useful for developing appropriate strategies for the prevention and treatment of type 2 diabetes. Accordingly, in the present study, we conducted OGTT in residents aged between 22 and 85 years in Koshi City, Kumamoto Prefecture, Japan, to assess the prevalence of diabetes and impaired glucose tolerance (IGT) and to clarify the characteristics of insulin secretion and insulin sensitivity in older individuals.

## 2. Materials and Methods

### 2.1. Study Design and Participants

Koshi City is a rural area in Kumamoto Prefecture, Japan. A flyer was distributed to all households (approximately 20,000 households) in Koshi City, informing them that a glucose tolerance test would be performed on the residents and inviting them to participate in the study. The following were excluded from the study on the basis of self-reported information: people being treated for diabetes; people with fasting blood glucose of 126 mg/dl or more or HbA1c of 6.5% or more on physical examination; and people who had undergone gastric resection. From August 2020 to August 2021, a total of 160 residents aged 22 to 85 years agreed to participate in the study and underwent an OGTT. This study was conducted in accordance with the Declaration of Helsinki. The study protocol was approved by the ethics committees of the Kumamoto University Graduate School of Medicine (No. 2602, 8 November 2022). Both written informed consent and opt-out informed consent were obtained from all participants.

### 2.2. Anthropometrics and Biochemical Measurements

A 75-g OGTT was carried out after an overnight fast (12–16 h) between 8:00 and 10:30 am. Blood samples were collected at 0, 30, 60, and 120 min. Based on plasma glucose levels, participants were categorized as having normal glucose tolerance (NGT; fasting plasma glucose [FPG] < 110 mg/dL and 2-h plasma glucose [2hPG] < 140 mg/dL), isolated impaired fasting glucose (IFG; FPG = 110–125 mg/dL and 2hPG < 140 mg/dL), IGT (FPG < 126 mg/dL and 2hPG = 140–199 mg/dL), and diabetes mellitus (FPG ≥ 126 mg/dL and/or 2hPG ≥ 200 mg/dL) according to World Health Organization criteria. Plasma glucose levels were measured by hexokinase enzymatic analysis, HbA1c levels were measured by high-performance liquid chromatography, and serum insulin levels were measured by chemiluminescent enzyme-linked immunosorbent assay. These measurements were conducted by BML, Inc. (Kumamoto, Japan).

Calf circumference, grip strength, and body composition were also measured by a qualified physician (K.F.) in the 88 individuals who agreed to participate. Calf circumference and grip strength were measured bilaterally, and the larger value was used for analysis. Skeletal muscle mass and body fat mass were measured by bioelectrical impedance analysis (BIA, InBody^®^ 270, Inbody Japan Co. Ltd., Tokyo, Japan). The skeletal muscle index was calculated as skeletal muscle mass/(height)^2^. Percent body fat was calculated as (body fat mass/body weight) × 100.

### 2.3. Calculation of Indices of Insulin Secretion and Insulin Sensitivity

HOMA-IR, an index of insulin resistance, was calculated as fasting serum insulin (μU/mL) × FPG (mg/dL)/405 [22]. The Matsuda index was calculated as {10,000/square root of (FPG [mg/dL] × fasting insulin [μU/mL] × mean glucose [mg/dL] × mean insulin during OGTT [μU/mL])} [23]. HOMA-β was calculated as (fasting insulin [μU/mL] × 360)/(FPG [mg/dL] − 63) [22]. The insulinogenic index was calculated as (insulin at 30 min − insulin at 0 min)/(glucose at 30 min − glucose at 0 min) [24]. The disposition index was calculated as the Matsuda index × (AUC-insulin/AUC-glucose), as previously reported [25].

### 2.4. Statistical Analysis

Data are presented as the mean ± standard deviation (SD). To approximate a normal distribution, HOMA-IR, Matsuda index, HOMA-β, insulinogenic index, and disposition index values were logarithmically transformed before statistical analysis. Differences in means were tested using an unpaired two-tailed Student’s *t*-test or one-way analysis of variance (ANOVA) with Tukey’s post hoc test. Frequencies were compared using the chi-square test. Pearson’s correlation coefficient analysis was performed to assess the relationship between two variables. Multivariate stepwise regression analysis was performed to explore the relationship of the indices of insulin secretion and insulin sensitivity with age, sex, body mass index (BMI), percent body fat, skeletal muscle index, calf circumference, and grip strength. A value of *p* < 0.05 was considered to be statistically significant. All statistical analyses were performed using Prism version 10 (GraphPad Software, Boston, MA, USA).

## 3. Results

### 3.1. Diabetes and Prediabetes Prevalence by Age Group

Participants were divided into four groups—group 1, 20–39 years; group 2, 40–59 years; group 3, 60–74 years; group 4, 75 years or older—according to age. The characteristics of each group are shown in Table 1. BMI was comparable among the four groups, while FPG was significantly higher in the two groups aged 60 years or older than in the two groups aged under 60 years.

Figure 1A shows the plasma glucose levels during the 75-g OGTT. The area under the curve (AUC) of plasma glucose during the OGTT (AUC-glucose) was significantly higher in the groups aged 60 years or older than in the groups aged under 60 years (Figure 1B), indicating that the older individuals were more glucose intolerant. Consistently, HbA1c was positively correlated with age (Figure 1C). The proportion of individuals with NGT decreased with age. On the other hand, the prevalence of newly diagnosed diabetes increased with age (0%, 2.0%, 15.5%, and 21.4% for groups 1–4, respectively). The prevalence of IGT also markedly increased with age (4.0%, 16.3%, 25.9%, and 42.9% for groups 1–4, respectively), whereas the prevalence of isolated IFG did not increase with age (Table 1 and Figure 1D). The prevalence of IGT tended to be higher in women than in men in younger age groups (aged 20–59 years) (Figure 1E,F), but the difference was not significant.

### 3.2. Relationship Between Age and Indices of Insulin Sensitivity

Increased body weight/adiposity and decreased muscle mass have been suggested to contribute to the reduced insulin sensitivity in older adults [2]. We thus investigated the relationship between aging and various parameters related to fat and muscle mass. No correlation between age and BMI was observed in the participants (Figure 2A). In terms of adiposity, there was a significant negative correlation between age and body fat mass (*r* = −0.28, *p* < 0.01) (Figure 2B). There was a nonsignificant tendency for the percentage of body fat to decrease with age (Figure 2C). Calf circumference (*r* = −0.22, *p* < 0.05) and grip strength (*r* = −0.37, *p* < 0.001) were negatively correlated with age (Figure 2D,E), and similar trends were observed for age with skeletal muscle mass (*r* = −0.18) and the skeletal muscle index (*r* = −0.18) (Figure 2F,G). A previous Japanese study reported a decrease in the HOMA-IR with age [19]. However, no correlation was found between age and indices of insulin action (HOMA-IR and Matsuda index) (Figure 2H,I), indicating that insulin sensitivity did not change with age in the participants.

### 3.3. Relationship Between Age and Indices of Insulin Secretion

We next investigated the relationship between age and the indices of insulin secretion. A significant negative correlation between age and HOMA-β was observed (*r* = −0.31, *p* < 0.0001) (Figure 3A). HOMA-β was significantly lower in groups 2–4 than in group 1 and in group 3 than in group 2 (group 1, 90.6 μU/mL/mg/dL; group 2, 64.0 μU/mL/mg/dL; group 3, 54.2 μU/mL/mg/dL; group 4, 58.9 μU/mL/mg/dL) (Figure 3B). The AUC of plasma insulin during the OGTT did not differ significantly among the age groups (Figure 3C,D). However, the insulinogenic index, a marker of early insulin secretion, declined with age (*r* = −0.33, *p* < 0.0001) and was significantly lower in the groups aged 60 years or older than in age group 1 (Figure 3E,F), suggesting that aging impairs early insulin secretion. The disposition index reflects the ability of β-cells to upregulate insulin secretion in response to a decrease in insulin sensitivity [25], and an age-dependent decline in the disposition index has been reported in Japanese adults between their 20s and 50s [26]. The disposition index was negatively correlated with age (*r* = −0.43, *p* < 0.0001), and the values were also significantly decreased in the groups aged 60 years or older than in the groups aged under 60 years (Figure 3G,H). Although HOMA-β, the insulinogenic index, and the disposition index decreased with age, these indices did not differ between groups 3 (60–74 years) and 4 (75 years or older) (Figure 3B,F,H). These findings may suggest that there is a limit to the decline in insulin secretion with age, or that the rate of the decline is slower in people over the age of 60.

### 3.4. Effects of Aging in People with NGT

Insulin secretion is impaired in individuals with IGT and type 2 diabetes [16], and the prevalence of both IGT and diabetes was higher in the older groups (Figure 1D). To rule out the possibility that the lower insulin secretion in the older participants was simply due to the high prevalence of IGT and type 2 diabetes, we examined the effects of aging in people with NGT. A significant positive correlation between age and HbA1c was still observed (Figure 4A). Because HOMA-β, the insulinogenic index, and the disposition index were reduced in the groups aged 60 years or older than in those aged under 60 years, people with NGT (n = 106) were divided into two groups (Table 2): 60 years or older (older group) and under 60 years (younger group). The AUC-glucose was significantly higher in the older group than in the younger group (Figure 4B,C). Consistent with the previous findings, HOMA-β, the insulinogenic index, and the disposition index were negatively correlated with age (*r* = −0.27 to −0.34) (Figure 4D–F). No correlation was found between age and the indices of insulin action (Figure 4G,H).

Previous studies showed that insulin secretion is associated with adiposity and skeletal muscle mass in Japanese people [27,28]. We next investigated the determinants of insulin secretion in people with NGT. HOMA-β was negatively associated with age (*r* = −0.34) and positively associated with BMI (*r* = 0.45), body fat mass (*r* = 0.36), percent body fat (*r* = 0.32), and calf circumference (r = 0.32) (Table 3). The insulinogenic index was negatively correlated with age (*r* = −0.35) and positively correlated with BMI (*r* = 0.25), body fat mass (*r* = 0.35), and percent body fat (*r* = 0.34). The disposition index showed a marginal negative correlation with age (*r* = −0.24) (Table 3). By multiple regression analysis, age and BMI were shown to be independent determinants of HOMA-β (Table 4). On the other hand, age was the only explanatory factor for the insulinogenic index and disposition index (Table 4).

## 4. Discussion

In the present study, we conducted OGTT in residents aged between 22 and 85 years in Koshi City, Kumamoto Prefecture, Japan, to assess the prevalence of diabetes and IGT. The prevalence of newly diagnosed diabetes was low (1 in 74: 1.4%) in individuals younger than 60 years but increased with age, peaking at 21.4% in the oldest age group (75 years or older). The prevalence of IGT, but not isolated IFG, also increased with age, and the peak prevalence of IGT exceeded 40% at the age of 75 years or older, indicating the importance of performing OGTT to detect glucose intolerance in older people. The high prevalence of type 2 diabetes and IGT in the older participants is consistent with reports from European and Asian countries [29,30].

Many previous studies have reported that insulin resistance increases with age [4,5,6,20], but we did not observe a worsening of insulin resistance with increasing age. It has been reported that insulin action, as measured by the euglycemic insulin clamp, decreases with age in Europeans, but that the age-related increase in insulin resistance is no longer statistically significant after adjustment for BMI [31]. We also found that BMI, but not age, is an independent determinant of HOMA-IR and the Matsuda index in NGT individuals by multiple regression analysis (Appendix A). Body fat accumulation is strongly associated with insulin resistance [2], and body fat mass did not increase with age in this study. The fact that BMI/adiposity did not increase with age may be one reason why insulin resistance was not worse in older people in this study. Further studies are needed in this regard.

In addition to obesity, the development of insulin resistance in older adults has been suggested to be associated with the loss of skeletal muscle mass [32,33]. While we observed a marginal correlation between age and the skeletal muscle index, multiple regression analysis did not show that the skeletal muscle index was a significant contributor to HOMA-IR and the Matsuda index in individuals with NGT (Appendix A). Similarly, no correlation between skeletal muscle mass and the Matsuda index was reported in another Japanese study [20]. These results may suggest that a reduction in skeletal muscle mass has less impact on insulin sensitivity in older people, at least in Japanese individuals with NGT.

Regarding insulin secretion in Japanese subjects, one study reported that HOMA-β declined with age [19], suggesting that insulin secretion decreases with age. On the other hand, another study showed no age-related change in the insulinogenic index in Japanese subjects after the age of 65 [20]. Insulin secretion in an OGTT was also comparable between middle-aged and older Japanese subjects [21]. Therefore, the effect of aging on insulin secretion in Japanese is controversial. In the present study, we found a significant negative correlation of age with HOMA-β, the insulinogenic index, and the disposition index in all participants and in those with NGT. In addition, age was shown to be the only independent determinant of these insulin secretion indices in a multiple regression analysis. Our results suggest that an age-related decline in insulin secretion plays an important role in the development of glucose intolerance in older Japanese adults. Chronic inflammation is a hallmark of aging, and the circulating levels of inflammatory cytokines and biomarkers (such as CRP) increase with age [34]. Because inflammation-related cytokines/adipokines such as IL-6, TNFα, and adiponectin have been reported to regulate insulin secretion [35,36,37], measuring these molecules would provide additional information on the age-related impairment of insulin secretion. The accumulation of senescent β-cells, another hallmark of aging, may also explain, at least in part, the age-related insulin secretion defect [38]. Senolysis (removal of senescent β-cells) has been reported to improve insulin secretion in mice [39].

In this study, the highest prevalence of type 2 diabetes and IGT was seen in the group aged over 75 years. However, we did not find a significant difference in insulin secretion between the groups aged 65–74 years and 75 years or older. The reasons why the highest frequency of glucose intolerance was observed in the group aged 75 years or older are unclear. Normal insulin secretion is pulsatile, but abnormalities in insulin pulsatility have been reported in older adults [1,2]. Further studies are needed to clarify the reasons for the high prevalence of glucose intolerance in the very old group.

There are several limitations to this study. First, because of the cross-sectional design, it is inappropriate to assess the cause–effect relationship and to extrapolate the age-related decline in insulin secretion for each individual. Second, we found the age-related decline in insulin secretion in Japanese subjects, but the number of participants was relatively small. Third, insulin secretion is affected by oxidative stress, endoplasmic reticulum stress, inflammation, and hypoxia [40]. However, we did not measure parameters related to these stresses. Fourth, we cannot exclude the possibility of selection bias in the study. We enrolled only non-diabetic individuals on the basis of self-reported information. It is therefore possible that people with increased insulin resistance were already diagnosed with diabetes and therefore excluded from the study. Finally, we did not assess the lifestyle of the participants, such as their dietary habits and physical activity.

## 5. Conclusions

In conclusion, this study showed that the age-related high prevalence of diabetes and IGT is associated with impaired insulin secretion rather than insulin resistance in Japanese individuals. Further studies are necessary to understand why insulin secretion declines with age in order to develop new strategies for the prevention and treatment of type 2 diabetes.

## Figures and Tables

**Figure 1 biomedicines-13-00380-f001:**
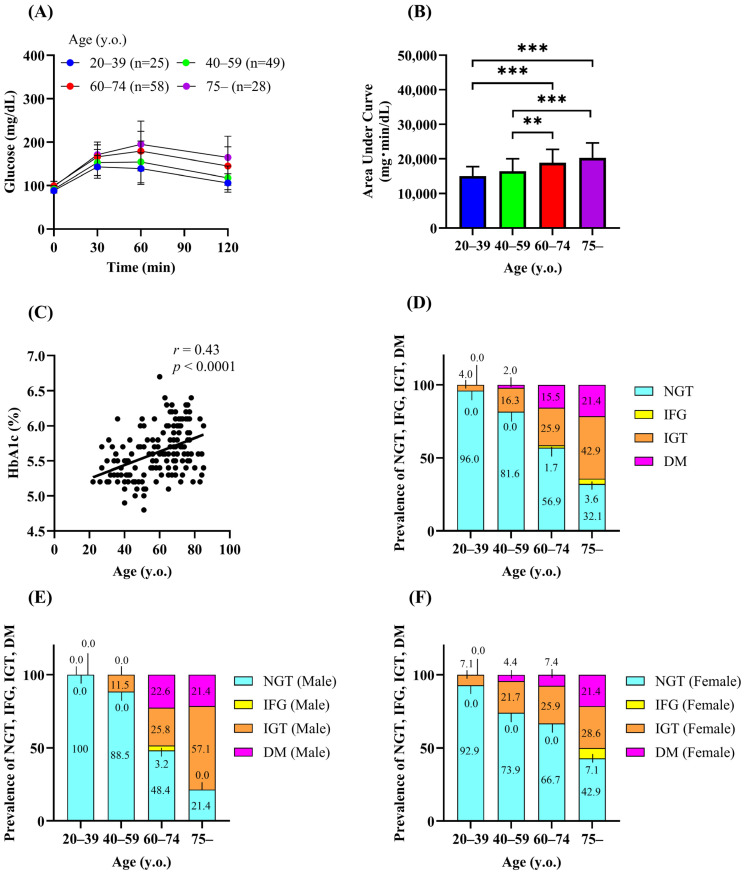
Prevalence of diabetes and impaired glucose tolerance in Koshi City. Plasma glucose (**A**) and the area under the curve of plasma glucose (**B**) during the 75-g OGTT. Correlation between age and HbA1c (**C**). Prevalence of normal glucose tolerance (NGT), impaired fasting glucose (IFG), impaired glucose tolerance (IGT), and diabetes (DM) (**D**–**F**). Data are shown as the mean ± SD. ** *p* < 0.01, *** *p* < 0.001.

**Figure 2 biomedicines-13-00380-f002:**
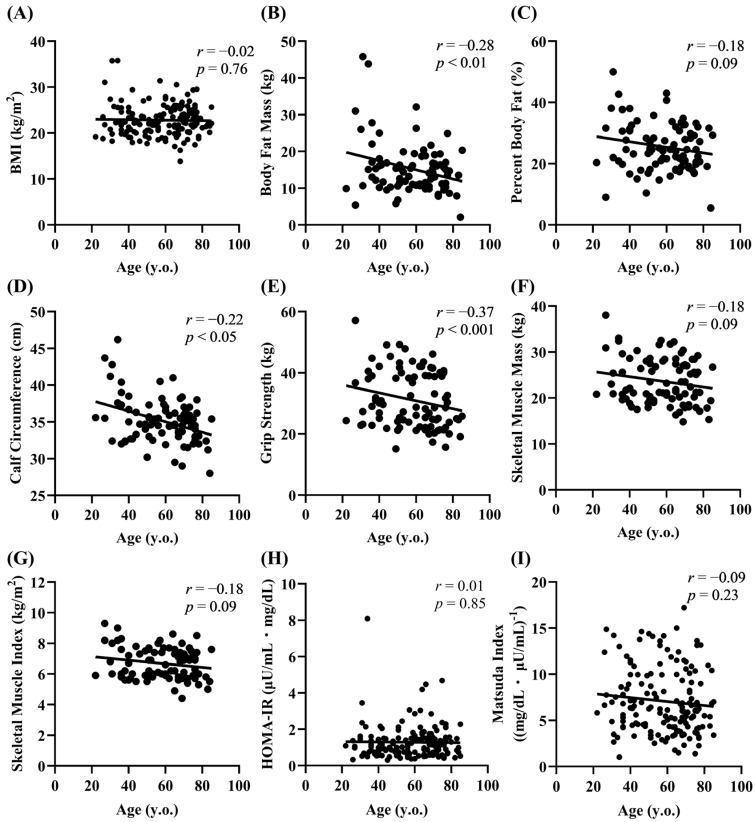
Relationship of age with fat/muscle mass parameters and the insulin sensitivity index. Correlation of age with adiposity (**A**–**C**), muscle mass (**D**–**G**), and the insulin sensitivity index (**H**,**I**).

**Figure 3 biomedicines-13-00380-f003:**
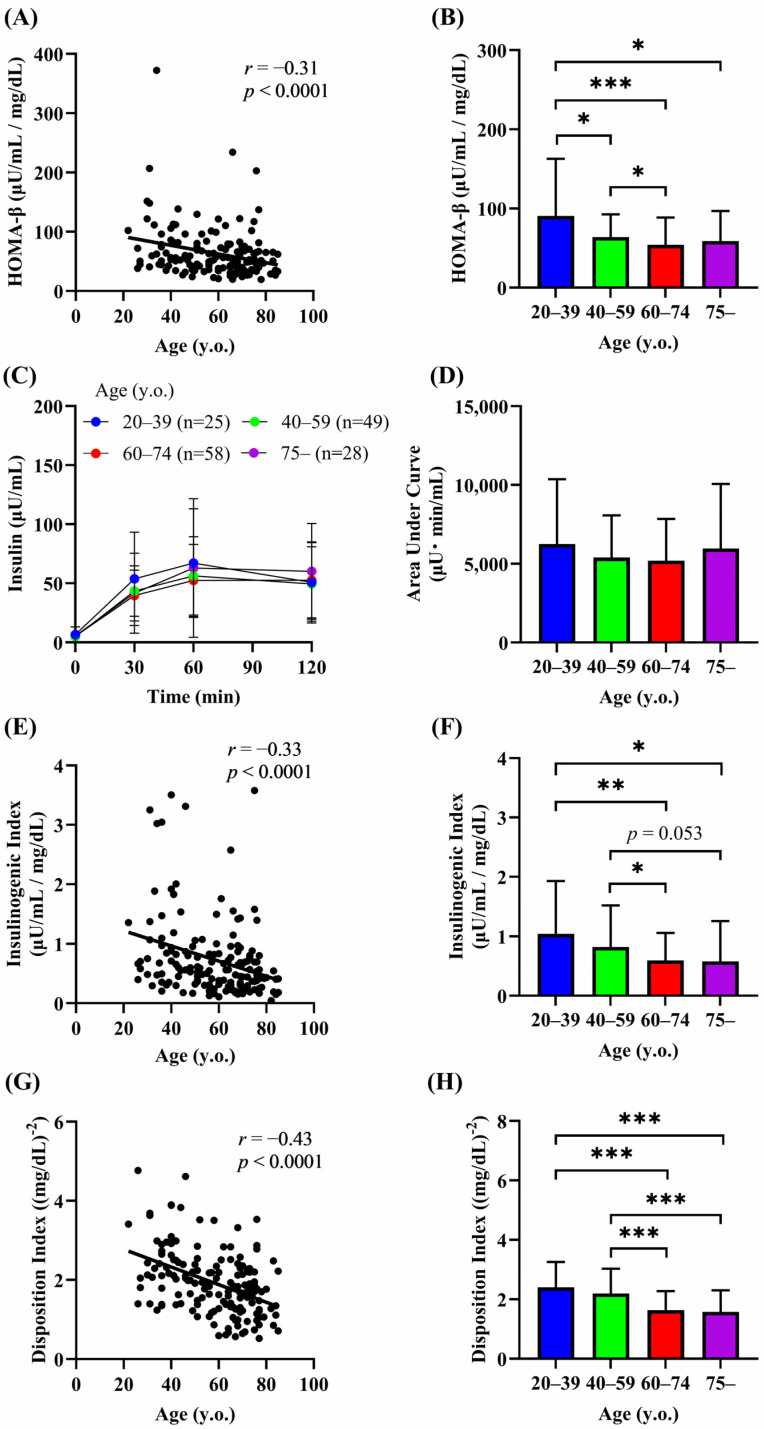
Relationship between age and indices of insulin secretion. Correlation of age with HOMA-β (**A**,**B**), insulin levels during the OGTT (**C**,**D**), insulinogenic index (**E**,**F**), and disposition index (**G**,**H**). Data are shown as the mean ± SD. * *p* < 0.05, ** *p* < 0.01, *** *p* < 0.01.

**Figure 4 biomedicines-13-00380-f004:**
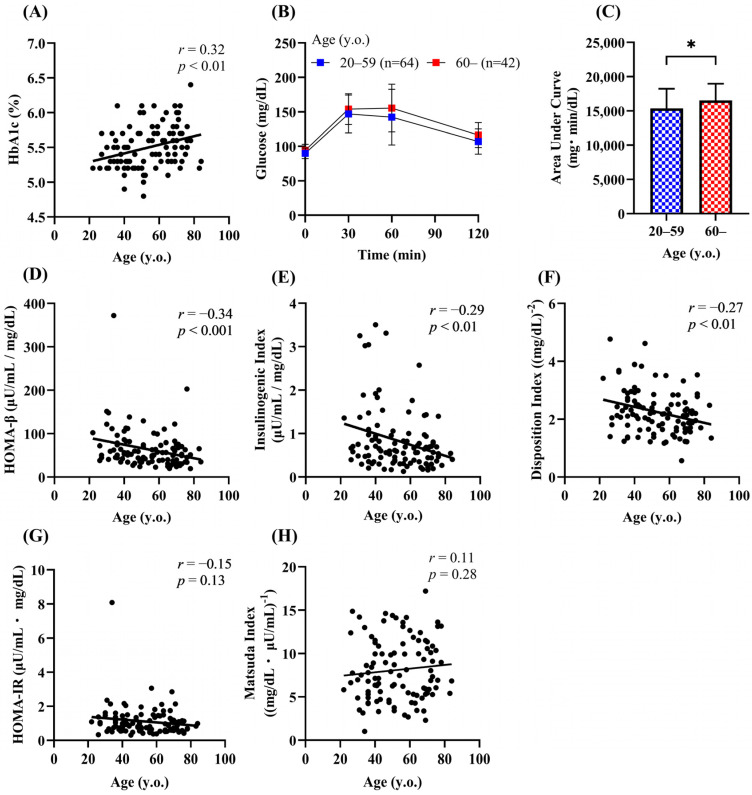
Effects of aging on glucose tolerance, insulin secretion, and insulin sensitivity in people with NGT. Correlation of age with the HbA1c (**A**), HOMA-β (**D**), insulinogenic index (**E**), disposition index (**F**), HOMA-IR (**G**), and Matsuda index (**H**) in the NGT group (n = 106). Plasma glucose (**B**) and the area under the curve of plasma glucose (**C**) during the 75-g OGTT. Data are shown as the mean ± SD. * *p* < 0.05.

**Table 1 biomedicines-13-00380-t001:** Characteristics of the individuals in the different age groups.

Groups	Group 1 (20–39)	Group 2 (40–59)	Group 3 (60–74)	Group 4 (75–)	*p*
n (M, F)	25 (11, 14)	49 (26, 23)	58 (31, 27)	28 (14, 14)	
Age (years)	32.6 ± 4.4	49.1 ± 6.2	67.6 ± 3.9	78.4 ± 3.4	
NGT, n (%)	24 (96.0)	40 (81.6)	33 (56.9)	9 (32.1)	
IFG, n (%)	0 (0)	0 (0)	1 (1.7)	1 (3.6)	
IGT, n (%)	1 (4.0)	8 (16.3)	15(25.9)	12(42.9)	
DM, n (%)	0 (0)	1 (2.0)	9 (15.5)	6 (21.4)	
BMI (kg/m^2^)	23.7 ± 4.8	22.2 ± 2.8	23.0 ± 3.3	22.9 ± 2.7	0.343
FPG (mg/dL)	88.6 ± 5.7	91.4 ± 8.5	99.4 ± 10.3 ***^, †††^	98.3 ± 12.2 **^, †^	<0.001
HbA1c (%)	5.43 ± 0.24	5.45 ± 0.32	5.76 ± 0.33 ***^, †††^	5.79 ± 0.37 ***^, †††^	<0.001
F-IRI (µU/mL)	6.62 ± 6.5	4.83 ± 2.3	5.29 ± 3.2	5.39 ± 3.1	0.259
HOMA-IR (μU/mL·mg/dL)	1.48 ± 1.53	1.11 ± 0.58	1.32 ± 0.84	1.35 ± 0.88	0.483
Matsuda Index ((mg/dL·μU/mL)^−1^)	7.25 ± 3.7	7.65 ± 3.6	6.84 ± 3.4	6.40 ± 3.4	0.403
HOMA-β (μU/mL/mg/dL)	90.6 ± 72	64.0 ± 29	54.2 ± 34 ***	58.9 ± 38 *	0.002
Insulinogenic Index (μU/mL/mg/dL)	1.05 ± 0.9	0.82 ± 0.7	0.59 ± 0.5 *	0.58 ± 0.7 *	0.006
Disposition Index ((mg/dL)^−2^)	2.40 ± 0.9	2.20 ± 0.8	1.64 ± 0.6 ***^, †††^	1.58 ± 0.7 ***^, ††^	<0.001
Body Fat Mass (kg) ^§^	21.1 ± 12.4	14.8 ± 5.7 *	14.0 ± 3.9 **	13.0 ± 5.5 **	0.006
Body Fat Percentage (%) ^§^	29.5 ± 11.0	25.5 ± 7.9	24.9 ± 6.1	23.3 ± 7.4	0.181
Skeletal Muscle Mass (kg) ^§^	25.8 ± 5.9	23.6 ± 4.5	23.2 ± 5.0	22.4 ± 4.4	0.276
Skeletal Muscle Index (kg/m^2^) ^§^	7.21 ± 1.3	6.65 ± 0.9	6.58 ± 1.0	6.49 ± 1.0	0.206
Calf Circumference (cm) ^§^	37.7 ± 4.7	35.3 ± 2.1 *	34.7 ± 2.5 **	33.5 ± 2.4 ***	0.001
Grip Strength (kg) ^§^	33.5 ± 9.9	32.8 ± 9.7	30.5 ± 9.1	26.8 ± 7.3	0.150

Data are shown as the mean ± SD. NGT, normal glucose tolerance; IFG, impaired fasting glucose; IGT, impaired glucose tolerance; DM, diabetes mellitus; BMI, body mass index; FPG, fasting plasma glucose; HbA1c, glycated hemoglobin (hemoglobin A1c); F-IRI, fasting immunoreactive insulin; HOMA-IR, homeostatic model assessment for insulin resistance; HOMA-β, homeostatic model assessment β-cell function. § Data were available for 88 of the participants (group 1, n = 14; group 2, n = 30; group 3, n = 29; and group 4, n = 15). * *p* < 0.05 vs. Group 1, ** *p* < 0.01 vs. Group 1, *** *p* < 0.001 vs. Group 1, † *p* < 0.05 vs. Group 2, †† *p* < 0.01 vs. Group 2, ††† *p* < 0.001 vs. Group 2.

**Table 2 biomedicines-13-00380-t002:** Characteristics of NGT individuals.

Groups	Younger Group (20–59)	Older Group (60–)	*p*
n (M, F)	64 (34, 30)	42 (18, 24)	
Age (years)	42.6 ± 9.6	70.1 ± 5.5	
BMI (kg/m^2^)	22.5 ± 3.5	22.4 ± 3.1	0.919
FPG (mg/dL)	89.9 ± 7.7	94.2 ± 8.7	0.009
HbA1c (%)	5.43 ± 0.29	5.59 ± 0.28	0.006
F-IRI (µU/mL)	5.21 ± 4.3	4.28 ± 2.1	0.194
HOMA-IR (μU/mL·mg/dL)	1.18 ± 1.03	1.00 ± 0.53	0.351
Matsuda Index ((mg/dL·μU/mL)^−1^)	7.97 ± 3.6	8.38 ± 3.5	0.474
HOMA-β (μU/mL/mg/dL)	71.4 ± 50	53.3 ± 33	0.006
Insulinogenic Index (μU/mL/mg/dL)	0.94 ± 0.8	0.65 ± 0.5	0.043
Disposition Index ((mg/dL)^−2^)	2.38 ± 0.8	2.03 ± 0.6	0.023
Body Fat Mass (kg) ^§^	15.6 ± 7.6	13.2 ± 4.4	0.158
Body Fat Percentage (%) ^§^	25.2 ± 8.2	23.2 ± 6.7	0.321
Skeletal Muscle Mass (kg) ^§^	24.6 ± 5.1	23.4 ± 4.7	0.343
Skeletal Muscle Index (kg/m^2^) ^§^	6.91 ± 1.0	6.49 ± 1.0	0.117
Calf Circumference (cm) ^§^	36.0 ± 3.3	34.6 ± 2.6	0.077
Grip Strength (kg) ^§^	34.4 ± 9.7	29.4 ± 8.6	0.042

Data are shown as the mean ± SD. BMI, body mass index; FPG, fasting plasma glucose; HbA1c, glycated hemoglobin (hemoglobin A1c); F-IRI, fasting immunoreactive insulin; HOMA-IR, homeostatic model assessment for insulin resistance; HOMA-β, homeostatic model assessment β-cell function. ^§^ Data were available for 63 of the participants (younger group, n = 38; older group, n = 25).

**Table 3 biomedicines-13-00380-t003:** Correlation coefficients of the relationship between indices of insulin secretion and various parameters.

	HOMA-β		Insulinogenic Index		Disposition Index	
	*r*	*p*	*r*	*p*	*r*	*p*
Age	−0.34	0.007	−0.35	0.004	−0.24	0.058
Sex	−0.17	0.187	−0.09	0.502	−0.21	0.096
BMI	0.45	0.0002	0.25	0.049	−0.23	0.073
Body Fat Mass	0.36	0.004	0.35	0.005	−0.08	0.518
Percent Body Fat	0.32	0.012	0.34	0.006	0.03	0.808
Skeletal Muscle Mass	−0.08	0.527	−0.06	0.656	−0.17	0.182
Skeletal Muscle Index	0.09	0.469	0.001	0.991	−0.23	0.076
Calf Circumference	0.32	0.010	0.15	0.256	−0.20	0.109
Grip Strength	0.03	0.818	0.001	0.992	−0.17	0.190

The Pearson correlation coefficient (*r*) and each probability value (*p*) are shown. Statistically significant results (*p* < 0.05) are shown in bold. Individuals with NGT (n = 106) are included in the analysis. Body composition data were available for 63 NGT individuals.

**Table 4 biomedicines-13-00380-t004:** Multiple regression analysis for indices of insulin secretion.

	HOMA-β		Insulinogenic Index		Disposition Index	
	Std β	*p*	Std β	*p*	Std β	*p*
Age	−0.064	0.043	−0.053	0.013	−0.028	0.040
Sex	−0.144	0.238	0.049	0.547	−0.015	0.771
BMI	0.144	0.011	0.056	0.134	−0.019	0.423
Percent Body Fat	−0.027	0.483	0.021	0.408	0.007	0.671
Skeletal Muscle Index	−0.064	0.433	−0.055	0.315	0.004	0.905
Calf Circumference	0.012	0.841	−0.018	0.655	−0.012	0.653
Grip Strength	0.041	0.417	0.006	0.855	−0.005	0.819

The standardized regression coefficient (Std β) and each probability value (*p*) are shown. Statistically significant results (*p* < 0.05) are shown in bold. Individuals with NGT (n = 106) are included in the analysis. Body composition data were available for 63 NGT individuals.

## Data Availability

The data that support the findings of this study is available on request to the corresponding author. The data are not publicly available due to privacy or ethical restrictions.

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
