# Peer review of "Age-Related Glucose Intolerance Is Associated with Impaired Insulin Secretion in Community-Dwelling Japanese Adults: The Kumamoto Koshi Study"

_biomedicines, 2025, doi:10.3390/biomedicines13020380_

Round 1
Reviewer 1 Report
Comments and Suggestions for Authors
It is a well conducted study useful in the clinical practice.
THe major question is why there is impaired insulin secretion.
To answer this question, it would have been better had the authors measured plasma CRP, IL-6, TNF and adiponectin.
Author Response
We thank the reviewer for stating that “It is a well conducted study useful in the clinical practice” and for his/her constructive comments. We have substantially revised our manuscript thanks to his/her valuable comments.
THe major question is why there is impaired insulin secretion. To answer this question, it would have been better had the authors measured plasma CRP, IL-6, TNF and adiponectin.
We thank the reviewer for pointing this out and agree that it is very important to clarify the reasons for the age-related impairment of insulin secretion. However, this would be a major undertaking and beyond the scope of this first study. Measurement of plasma CRP, IL-6, TNFa, and adiponectin would provide additional information on the age-related insulin secretion defect, but we did not have the opportunity to measure the plasma levels of these molecules. Therefore, we have revised part of the Discussion section as follows: “Chronic inflammation is a hallmark of aging, and the circulating levels of inflammatory cytokines and biomarkers (such as CRP) increase with age [34]. Because inflammation-related cytokines/adipokines, such as IL-6, TNFα, and adiponectin, have been reported to regulate insulin secretion [35-37], measurement of these molecules would provide additional information on the age-related impairment of insulin secretion.” (lines 293 to 298).
Reviewer 2 Report
Comments and Suggestions for Authors
This study investigated the age-related decline in glucose tolerance and its association with insulin secretion and sensitivity in Japanese adults through an oral glucose tolerance test (OGTT). The results revealed a significant decrease in insulin secretion indices, including HOMA-β, insulinogenic index, and disposition index, with advancing age, demonstrating reductions of 25–30% in individuals aged 60 years and older compared to those under 40. Notably, age was identified as the primary explanatory factor for the decline in insulin secretion, independent of BMI or muscle mass, suggesting a direct impact of aging on pancreatic β-cell function. While no significant age-related changes in insulin sensitivity indices (HOMA-IR and Matsuda index) were observed, the prevalence of impaired glucose tolerance (IGT) increased markedly, reaching 42.9% in individuals aged 75 years and older. These findings highlight the critical role of aging in the progressive deterioration of insulin secretion, emphasizing the need for targeted interventions to mitigate age-related glucose intolerance and reduce the risk of type 2 diabetes in older populations.
However, I therefore have to point out some comments:
Line 3: The title "Japanese adults" lacks specificity. Consider specifying "community-dwelling" or "non-diabetic" adults for clarity and relevance.
Line 16: The description of the OGTT method is insufficient. Include details about participant preparation, such as fasting duration, to enhance reproducibility.
Line 18: The mention of "HOMA-IR or the Matsuda index" lacks context. Provide a brief definition of these indices on their first appearance to aid readers unfamiliar with these terms.
Lines 27-29: The claim of "diabetes prevalence exceeding 20%" lacks a supporting citation.
Line 35: Contradictory claims about whether aging increases or decreases insulin secretion are presented without clear resolution.
Lines 43-44: The reference to "a study reporting HOMA-IR decline with age" lacks a direct link to the context of the current study population.
Line 64: The recruitment period (August 2020–August 2021) is provided, but no information on recruitment rate or sample size justification is included.
Lines 66-67: The ethics approval number (No. 2602) is given without a specific date.
Line 78: The methods for measuring HbA1c and glucose lack details about assay types or laboratory accreditation, which are crucial for reproducibility.
Line 87: The formula for HOMA-IR is included, but no explanation is provided for its clinical relevance.
Line 103: The description of multivariate regression analysis omits which variables were included.
Lines 151-152: The text states a "negative correlation between age and body fat mass," but the clinical significance of this finding is not discussed.
Line 167: The statement "HOMA-β was significantly lower in groups 2–4" is vague. Quantify the differences to enhance clarity.
Lines 288-289: The authors note the cross-sectional design as a limitation but do not discuss how this affects the interpretation of causality.
Lines 293-294: The potential selection bias in recruitment is mentioned but not explored in depth.
Lines 297-298: The conclusion emphasizes insulin secretion decline with age but does not connect this sufficiently to preventive strategies for diabetes.
Author Response
We thank the reviewer for his/her constructive comments. We have substantially revised our manuscript based on his/her valuable comments.
Line 3: The title "Japanese adults" lacks specificity. Consider specifying "community-dwelling" or "non-diabetic" adults for clarity and relevance.
We thank the reviewer for pointing this out. In accordance with the reviewer's comment, we have changed the title of the revised manuscript to “Age-related glucose intolerance is associated with impaired insulin secretion in community-dwelling Japanese adults: The Kumamoto Koshi Study”.
Line 16: The description of the OGTT method is insufficient. Include details about participant preparation, such as fasting duration, to enhance reproducibility.
We appreciate the reviewer’s comment. We have revised the Abstract as follows: “Participants were recruited using a flyer, and the OGTT was performed after an overnight fast (12–16 h) between 8:00 and 10:30 am.” (lines 18 to 20). We have also revised part of the Materials and Methods as follows: “A flyer was distributed to all households (approximately 20,000 households) in Koshi City, informing them that a glucose tolerance test would be performed on the residents and inviting them to participate in the study.” (lines 63 to 66).
Line 18: The mention of "HOMA-IR or the Matsuda index" lacks context. Provide a brief definition of these indices on their first appearance to aid readers unfamiliar with these terms.
We thank the reviewer for pointing this out. To the Abstract of our revised manuscript, we have added “HOMA-IR and the Matsuda index are indices of insulin action.” to explain the HOMA-IR and Matsuda index. (lines 20 to 21).
Lines 27-29: The claim of "diabetes prevalence exceeding 20%" lacks a supporting citation.
In the original manuscript, we stated that the global prevalence of diabetes has been reported to exceed 20% among adults aged 65–95 years, according to reference 3. In the revised manuscript, we have changed the sentence as follows: “the total diabetes prevalence exceeds 20% in every age group between 65 and 95 years worldwide [3].”
Line 35: Contradictory claims about whether aging increases or decreases insulin secretion are presented without clear resolution.
We thank the reviewer for pointing out this problem. We have revised part of the Introduction as follows: “Previous studies have been inconsistent, reporting that aging increases [6, 10], decreases [4, 7, 11-13], or does not change [5, 14, 15] insulin secretion. This variability may be due to confounding factors associated with aging, such as obesity and insulin resistance [1].” (lines 39 to 40).
Lines 43-44: The reference to "a study reporting HOMA-IR decline with age" lacks a direct link to the context of the current study population.
We thank the reviewer for highlighting this issue. We have revised part of the Results as follows: “A previous Japanese study reported a decrease in the HOMA-IR with age [19]. However, no correlation was found between age and indices of insulin action (HOMA-IR and Matsuda index) (Figure 2H and 2I), indicating that insulin sensitivity did not change with age in the participants.” (lines 167 to 168).
Line 64: The recruitment period (August 2020–August 2021) is provided, but no information on recruitment rate or sample size justification is included.
We thank the reviewer for pointing this out. We have revised part of the Materials and Methods as follows: “Koshi City is a rural area in Kumamoto Prefecture, Japan. A flyer was distributed to all households (approximately 20,000 households) in Koshi City, informing them that a glucose tolerance test would be performed on the residents and inviting them to participate in the study.” (lines 63 to 66).
Lines 66-67: The ethics approval number (No. 2602) is given without a specific date.
We apologize for the inadequate description in the original manuscript. We have revised the text as “(No. 2602, November 8, 2022)” (line 73). We have also revised part of the Institutional Review Board statement (line 342).
Line 78: The methods for measuring HbA1c and glucose lack details about assay types or laboratory accreditation, which are crucial for reproducibility.
We thank the reviewer for pointing this out. We have revised the manuscript as follows: “Plasma glucose levels were measured by hexokinase enzymatic analysis, HbA1c levels were measured by high-performance liquid chromatography, and serum insulin levels were measured by chemiluminescent enzyme-linked immunosorbent assay. These measurements were conducted by BML, Inc. (Japan).” (lines 83 to 87). BML is an accredited laboratory.
Line 87: The formula for HOMA-IR is included, but no explanation is provided for its clinical relevance.
We appreciate the reviewer’s comment. We have revised the manuscript as follows: “HOMA-IR, an index of insulin resistance, was calculated as fasting serum insulin (μU/mL) × FPG (mg/dL) / 405 [22].” (line 97).
Line 103: The description of multivariate regression analysis omits which variables were included.
We thank the reviewer for pointing this out. We have revised the manuscript as follows: “Multivariate stepwise regression analysis was performed to explore the relationship of the indices of insulin secretion and insulin sensitivity with age, sex, body mass index (BMI), percent body fat, skeletal muscle index, calf circumference, and grip strength.” (lines 112 to 115).
Lines 151-152: The text states a "negative correlation between age and body fat mass," but the clinical significance of this finding is not discussed.
We appreciate the reviewer’s insightful comment. We have revised part of the Discussion as follows: “Body fat accumulation is strongly associated with insulin resistance [2], and body fat mass did not increase with age in this study. The fact that BMI/adiposity did not increase with age may be one reason why insulin resistance was not worse in older people in this study. Further studies are needed in this regard.” (lines 269 to 271).
Line 167: The statement “HOMA-β was significantly lower in groups 2–4” is vague. Quantify the differences to enhance clarity.
We apologize for the poor description in the original manuscript. We have revised the text as follows: “A significant negative correlation between age and HOMA-β was observed (r = −0.31, p < 0.0001) (Figure 3A). HOMA-β was significantly lower in groups 2–4 than in group 1 and in group 3 than in group 2 (group 1, 90.6 μU/mL / mg/dL; group 2, 64.0 μU/mL / mg/dL; group 3, 54.2 μU/mL / mg/dL; group 4, 58.9 μU/mL / mg/dL) (Figure 3B).” (lines 179 to 181).
Thanks to your comment, we realized that the HOMA-b unit was incorrectly described as mg/dL / mU/mL in the original manuscript. We have changed it to the correct unit (mU/mL / mg/dL) in the revised manuscript. Please note that this correction is a unit error and the HOMA-b values themselves remain unchanged. We apologize for the error.
Lines 288-289: The authors note the cross-sectional design as a limitation but do not discuss how this affects the interpretation of causality.
We thank the reviewer for pointing this out. We have revised the text as follows: “First, because of the cross-sectional design, it is inappropriate to assess the cause–effect relationship and to extrapolate the age-related decline in insulin secretion to each individual.” (line 310).
Lines 293-294: The potential selection bias in recruitment is mentioned but not explored in depth.
We appreciate the reviewer’s useful comment. We have revised the text as follows: “Fourth, we cannot exclude the possibility of selection bias in the study. We enrolled only non-diabetic individuals on the basis of self-reported information. It is therefore possible that people with increased insulin resistance were already diagnosed with diabetes and therefore excluded from the study.” (lines 315 to 318).
Lines 297-298: The conclusion emphasizes insulin secretion decline with age but does not connect this sufficiently to preventive strategies for diabetes.
We thank the reviewer for this comment on our original manuscript and are in agreement. In the revised manuscript, we have added information to the Discussion stating that there is a report that removal of senescent b-cells (senolysis) in mice can ameliorate age-related insulin secretion defects (lines 298 to 301). We have also revised the text in the Conclusion as follows: “Further studies are necessary to understand why insulin secretion declines with age in order to develop new strategies for the prevention and treatment of type 2 diabetes.” (lines 323 to 325).
Round 2
Reviewer 2 Report
Comments and Suggestions for Authors
no